

# Interdecadal shift in the impact of winter land-sea thermal contrasts on following spring transcontinental dust transport pathways in North Africa

Qi Wen [1], Yan Li[1, *], Mengying Du[1], Wenjun Song[1], Linbo Wei[1], Zhilan Wang[2,3], Xu Li [1]

*1. Key Laboratory for Semi-Arid Climate Change of the Ministry of Education, College of Atmospheric Sciences, Lanzhou University, Lanzhou 730000, China*

*2. Key Laboratory of Water Security and Water Environment Protection in Plateau Intersection, Ministry of Education, College of Chemistry and Chemical Engineering, Northwest Normal University, Lanzhou 730070, PR China;*

*3. Lanzhou Institute of Arid Meteorology,China Meteorological Administration,Key Laboratory of Arid Climatic Change and Reducing Disaster of Gansu,Key Laboratory of Arid Climatic Change and Disaster Reduction of CMA,Lanzhou 730020,China*

Corresponding author: Yan Li (E-mail: liyanlz@lzu.edu.cn)

## Abstract

North Africa, the largest and most active dust source region globally, plays a critical role in the Earth's environment by dispersing dust over remote areas, especially in terms of circum-global transport that occurred many times since the 21st century. As a key indicator of the thermodynamic structure and dynamical circulation of the troposphere, the land-sea thermal contrast (LSC) could influence the variability of dust and subsequent large-scale propagation, but the extent of such influence is still unknown. This study reveals that around the late 1990s, the influence of pre-winter LSC on the spring dust transport pathway is reversed in North Africa, which is attributed to the bridging effect of the North Atlantic Oscillation (NAO). Before 2000, the warm land-cold ocean (+WLCO) pattern in pre-winter is typically associated with the NAO+ mode, and the anomalous northeasterly and zonal circulation in the following spring facilitate the westward transport of dust from the lower troposphere in West North Africa towards the Atlantic. After 2000, the reversed zonal temperature pattern (−WLCO) leads to the NAO− mode and enhances mid-latitude westerlies in winter, which persists into the next spring. Under conditions of unusually dry soil and strong dry convection, dust is mixed into the mid-to-upper troposphere and subsequently transported eastward globally, affecting regions including West Asia, northern China, the Pacific, and southeastern North America after 2000. This study underscores the critical role of sea-



land-atmosphere interaction in circum-global dust propagation and offers new
perspectives for investigating dust changes mechanism in the context of climate change.

# 1 Introduction

North Africa is one of the major sources of dust in the world (Engelstaedter et al.,
2006; Huneeus et al., 2011), and the long-range transport of dust has profound impacts
on Atlantic hurricanes (Sun et al., 2008; Rousseau-Rizzi and Emanuel, 2022), global
climate change (Westphal et al., 1987; Sassen et al., 2003; Kok et al., 2023), the carbon
cycle (Keith et al., 2006; Swap et al., 1992; Guieu et al., 2002), and human health
(Mallone et al., 2011; Brauer et al., 2012; Wang et al., 2020).

Under the amplified influence of global warming, North African dust activity has
experienced significant modifications in recent decades. Pronounced alterations in
large-scale atmospheric circulations, particularly the Hadley circulation and mid-
latitude westerlies (Feng et al., 2018; Cheng et al., 2022; Toggweiler, 2009; Abell et al.,
2021), have fundamentally reshaped dust transport patterns. Observational records
from 1980 to 2020 reveal divergent trends in regional dust export: a decreasing flux
toward the Atlantic ($-0.29 \pm 0.16\%$ decade$^{-1}$) contrasted by increasing Mediterranean
transport ($0.24 \pm 0.18\%$ decade$^{-1}$), which potentially associated with the Hadley cell's
expansion (Adame et al., 2022). Correspondingly, emerging evidence points to
increased frequency of extreme transcontinental dust events, exemplified by the June
2020 "Godzilla" dust storm that transported $24 \pm 3.2$ Tg of Saharan material circum-
globally through an "express lane"—mid-latitude westerly wind (Bi et al., 2024; Francis
et al., 2020; Asutosh et al., 2022). The dominant factors of significant decadal changes
in the propagation of dust in North Africa deserve further exploration.

Global warming has exhibited significant temporal and spatial heterogeneity. The
warming trend accelerated until the late 1990s, followed by a period of apparent
stagnation (Fyfe et al., 2013). This warming pattern has been particularly evident in
terrestrial regions compared to oceanic areas, known as terrestrial amplification (TA)
(Seltzer et al., 2023; Sutton et al., 2007; Byrne and O'Gorman, 2018). The TA effect
alters the magnitude of the land-sea thermal contrast (LSC) (Joshi et al., 2008; Byrne
and O'Gorman, 2013), which plays a critical role in regulating the climate system's
energy balance and redistribution, thereby altering the planetary wave patterns
throughout the entire troposphere (Held and Ting, 1990; Garfinkel et al., 2020). For
instance, the strong land-sea temperature gradient between the eastern coasts of Asia
and North America are prominent sources of baroclinicity, triggering eastward-
extending storm tracks, which in turn, energetically support the jet streams (Hoskins et
al., 1990; Brayshaw et al., 2009). As global warming intensifies, changes in the LSC
have substantially influenced key climate patterns, such as the intensity of monsoon
systems (Torres-Alavez et al., 2014; Tao et al., 2016; Roxy et al., 2015), the frequency
of tropical and Arctic cyclones (Tang et al., 2019; Day et al., 2018), and perturbations



in the westerly belt (He et al., 2014, 2018; Portal et al., et al., 2022), all of which could
exacerbate the frequency of extreme weather events.
In fact, midlatitude LSC plays a crucial role in interannual to interdecadal
atmospheric variability, potentially influencing North African dust transport pathways,
particularly circum-global circulation processes. According to the thermal-equilibration
theory, the asymmetry of the zonal surface temperature pattern can induce a global-
scale wave-like thermal structure, thereby triggering a resonance between the mid-
latitude circulation and temperature structure, or a zonal flow pattern (Marshall and SO,
1990; Mitchell and Derome, 1983). LSC variations, exemplified by the winter cold
ocean-warm land (COWL) pattern, are often closely linked to the North Atlantic
Oscillation (NAO) through tropospheric planetary wave modulation (Molteni et al.,
2011). The alternating phases of the NAO significantly affect the emission and
propagation of Saharan dust. Especially, during the NAO+ (NAO−) phase, dust is
typically transported westward (eastward) into the tropical Atlantic (eastern
Mediterranean) by northeasterly (southwesterly) winds (Moulin et al., 1997; Chiapello
et al., 1997; Ginoux et al., 2004; Riemer et al., 2006; Doherty et al., 2008; Kaskaoutis
et al., 2019; Dai et al., 2022). Subsequently, the dominant easterly transport of mineral
dust is further enhanced by the westerly jet stream, facilitating circum-global dust
dispersion and significantly influencing downstream regions such as Asia (Pu et al.,
2016; Liu et al.,2022; Awad et al., 2014).
Reanalyzed data and models results have demonstrated that the LSC have induced
significant modifications in planetary-scale atmospheric wave patterns (He et al., 2014,
2018), with the dominant airflow and dust transport pathways in North Africa being
affected. However, has the LSC affected the decadal variation of dust in North Africa?
And what is its impact mechanism? These issues have not been answered yet. We find
that a regime shift in dust transport dynamics around the late 1990s. During the pre-
2000 epoch, the COWL pattern, driven by land warming in pre-winter, is shown to
affect the westward transport path of North Africa dust during the subsequent spring.
After this period, reversed zonal temperature pattern (warm ocean and cold land,
WOCL) continues to enhance the eastward dust transport, facilitating circum-global
dispersion as far as southeastern North America. In addition, the mechanisms
underlying the trans-seasonal effects of this large-scale dynamical precursor signal and
its transport have been thoroughly elucidated.

## 2 Methods and data

### 2.1 Methods

#### 2.1.1 SVD of extratropical SAT and North African dust

Surface thermal modes have a significant impact on the alternation of the two
possible dynamic equilibria (wave or band components), which may affect North





African dust activity. The Singular Value Decomposition (SVD) analysis was
conducted to initially explore the relationship by examining the covariance matrices of
springtime North African dust concentration and pre-winter extratropical temperatures.
**2.1.2 The land–sea contrasts (LSC) index**
Firstly, the anomaly pattern associated with the 'traditional', empirically based
northern extratropical low-frequency variability is presented. This is characterized by
an EOF associated with the second principal component of the 500 hPa height (Z500)
anomaly in the northern hemisphere extratropic (20–80°N), which displays a
pronounced zonal asymmetry (Fig. 1). Molteni et al. (2011) defined the land-sea
contrast as the bandwave-2 component of the net surface heat flux, averaged over four
sectors of 90° longitude each. Given that the latent heat is approximately zero during
the winter months, it is sufficient to consider the difference in sensible heat between the
land and ocean surfaces. Therefore, referring to the approach of He et al. (2014), LSC
index (LSCI) can thus be expressed in a straightforward manner as the land-ocean
contrast of the SAT anomaly in the critical zone (east coast of North America and east
coast of East Asia) with the following equation.
$$LSCI = (SAT_{anom_A} - SAT_{anom_B}) + (SAT_{anom_C} - SAT_{anom_D}) \quad (1)$$

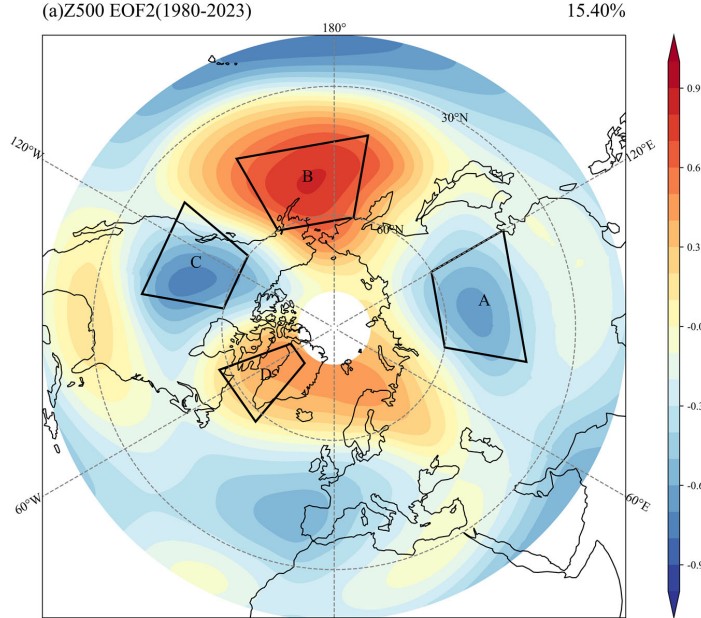


**Fig. 1: The second EOF of DJF mean 500 hPa geopotential height (Z500) during 1980–2023, with reference to**
**He et al. (2014).** The regions A, B, C and D represent the East Asian (40°N–60°N; 80°E–120°E), Pacific (40°N–



60°N; 170°E–150°W), North American (40°N–60°N; 130°W–100°W) and Greenland seas (57.5°N–77.5°N;
70°W–40°W), respectively.
The heat capacity of the land is considerably less than that of the oceans, resulting
in a significantly greater warming of the continents during winter compared to the
oceans under global warming. Consequently, a positive LSC value indicates a warmer
climate with a reduced temperature gradient between land and sea. During the winter
months, the anomalous warming of the land results in a shift from a negative to a
positive LSC signifying a reduction in the temperature disparity between the land and
the sea.

### 2.1.3 Two periods of determination

In accordance with established climatological standards, normal values are
typically calculated for a minimum of 30 consecutive years in order to obtain a
meaningful mean. As our study is concerned with inter-decadal climate change, an
analysis of shorter periods may yield different trends than those observed for longer
climatic periods. However, given that the MERRA-2 dust data only commence in 1980,
a compromise was reached. It is also considered appropriate to stratify the whole study
period into a first period (1980−2000) and a second period (2001−2023), reflecting the
different climatology phases, as can be seen in the results of the sliding t-test (Figure is
omitted). This temporal stratification is selected to analyses the interdecadal impact of
the LSC on dust transport in North Africa. In this study, composite, correlation and
regression are used to measure the relationship between dust and LSC, and the
significance test is based on the two-sided Student's t-test.

### 2.1.4 Selection of years for composite analysis in the two periods

Further investigation into the LSC-related dust transport characteristics in North
Africa during these two periods is conducted through composite analyses. The onefold
standard deviation of the standardized LSCI serve as thresholds for selection, with the
year's corresponding to the positive and negative phases of the LSC (Table 1).
Note that the composite analysis for the first period uses high value years (LSCI >
1) minus low value years (LSCI < −1), whereas the second period uses low value years
(LSCI < −1) minus high value years (LSCI > 1) minus low value years, which is related
to the interdecadal shift in the relationship between the winter LSC and spring dust in
North Africa.





**Table 1.** List of Year selection for composite analysis in this study.

| | First period (1980-2000) | | Second period (2001-2023) |
|---|---|---|---|
| LSCI>1 | 1983 | LSC>1 | 2002 |
| | 1987 | | 2015 |
| | 1989 | | 2016 |
| | 1993 | | |
| LSC<−1 | 1980 | LSC<−1 | 2010 |
| | 1982 | | 2011 |
| | 1985 | | 2013 |
| | 1996 | | 2021 |

## 2.2 Data

### 2.2.1 MERRA-2 global dust and atmospheric reanalysis

In order to obtain the longest possible dust sequence for study of relevant inter-decadal variability, the MERRA-2 dust data are selected here. The MERRA-2 dataset is a reanalysis product developed using the Goddard Earth Observing System of Systems (GEOS-5.12.4) atmospheric model, which simulates global aerosol properties using the radiatively coupled Goddard Chemistry, Aerosol, Radiation, and Transport (GOCART) model. MERRA-2 directly assimilates the aerosol optical depths derived from AERONET and MISR. MERRA-2 directly assimilates aerosol optical depths derived from the AERONET and MISR instruments, as well as bias-corrected dust concentrations and aerosol data from the Advanced Very High-Resolution Radiometer (AVHRR) and Moderate Resolution Imaging Spectroradiometer (MODIS) instruments. In the present study, monthly dust properties are considered, namely dust column mass density and meteorological and land conditions related to dust activities, including Z500, U200, UV500, PV, UV10, T2M, SM, etc., at a spatial resolution of 0.625° × 0.5°.

### 2.2.2 HadCRU5 global surface air temperature observations

The monthly SAT used to calculate the LSC index are derived from the Met Office Hadley Centre's observational dataset HadCRUT5. This is one of the main datasets used to monitor global and regional surface temperature variations and trends.

### 2.2.3 CMIP6 data

In order to investigate the impact of LSC on dust transport in North Africa, a comparison was made between historical simulations (1980−2014) from the 14 participating Coupled Model Intercomparison Project Phase 6 (CMIP6) models that contain both dust and meteorological information. The selected models are detailed in Table S1. Monthly outputs from CMIP6 are employed to examine the response of dust aerosols and upper zonal winds to the land-sea contrast in the model since the 1980s.





## 3 Result

### 3.1 Interdecadal LSC signal in pre-winter leads to change of the circum-global transport path of North Africa in the following spring

Utilizing the SVD analysis (see "Methods"), coherence is observed between pre-winter extratropical surface air temperature (SAT) in the Northern Hemisphere (NH) and spring dust mass column density (hereafter referred to as DUST) in North Africa. The first mode explains 43.15% of the total variance, and substantial correlation of R = 0.64 (11-year filtered correlation, R = 0.86) is demonstrated by the time series of the two variables (PC1-DUST and PC1-SAT). The spatial pattern of DUST field is revealed to follow a zonal tri-pole mode (Fig. 2a), with an interdecadal abrupt change around 2000 (Fig. 2c), which is consistent with the findings of Shi et al. (2021). On the other hand, the extratropical SAT field highlights the thermal contrast, with an opposite signal between Asia (Siberia) and the eastern Pacific, and an even greater thermal contrast between North America and the Greenland Sea (Fig. 2b). This spatial temperature pattern, called as the COWL, has also been found in previous studies (Wallace et al., 1996; Wu et al., 2004; He et al., 2014). Based on the east coast of North America and east coast of East Asia, LSCI is defined (see "Methods"), which shows a significant correlation of 0.86 with the PC1-SAT. The interannual variation of the LSCI (Fig. 2d) is consistent with the two phases of warming (warming acceleration and warming stagnation). The subsequent study will use this index to further analyze decadal variability.

The correlation between the pre-winter LSCI and the following spring PC1-DUST exhibits a stepwise change over time, with a stable and significant relationship between the two variables emerging only after the late 1990s (Fig. 2e). To further analyze the decadal impact of LSC on North African dust, we examine the regression spatial field of spring dust with respect to the pre-winter LSCI during the two periods (1980−2000 and 2001−2023; see "Methods"). Prior to 2000, significant positive regression coefficients are found in a small region of West Africa, while the relationship in the central region is not significant (Fig. 2f). After 2000, distinct negative regression coefficients are observed in the central region (Fig. 2g).



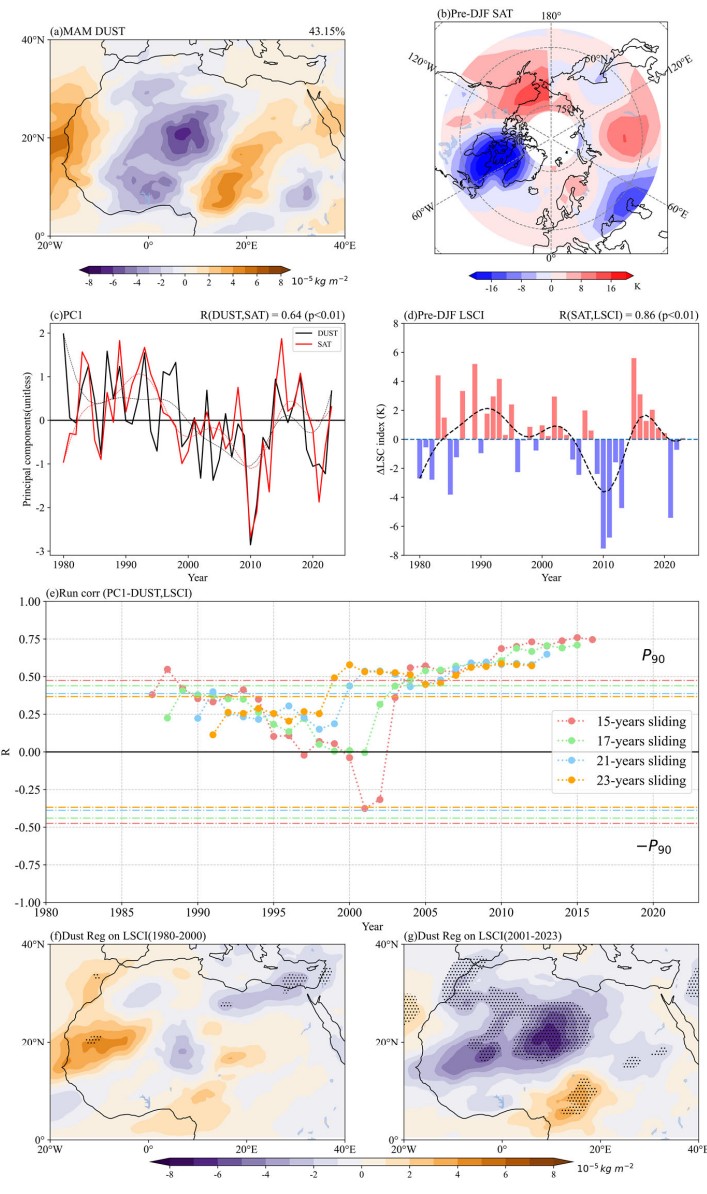



**Fig. 2: The relationship between pre-winter land-Sea thermal contrasts (LSC) and spring North African dust.** The SVD first mode between detrended spring (MAM) MERRA-2 dust mass column density (DUST) in North Africa (a) and pre-winter (DJF) surface air temperature (SAT) (b) from 1980 to 2023. (c) Time series (solid) and low-pass filtered (dashed) SVD first mode coefficients of the DUST (left, black line) and the extratropical SAT (right, red line). (d) LSC index time series and its corresponding low-pass filter (black dashed line), unit: K. (e) The sliding correlations between DJF LSC index and PC1-dust index under different moving windows (15, 17, 21, 23 years). The regression patterns of detrended DUST onto LSCI (standardization) during (f) 1980−2000 and (g) 2001−2023 (shading; $10^{-5}kg/m^2$). The dashed lines and dots indicate that the correlation coefficients pass the 90% confidence test.

In composite analysis, when warming in North America and East Asia alongside cooling along their eastern coasts (Fig. S1a), i.e., the LSC positive phase, positive dust anomalies over West Africa predominantly follow westward trajectories to the Atlantic Ocean during 1980-2000 (Fig. 3a). This westward transport pattern aligns with observations by Evan et al., who documented peak Atlantic dust export in the 1980s followed by a marked post-2000 decline (Evan et al., 2016). After 2000, dust related to negative LSC phase exhibit preferential eastward transport to West Asia and northern China via the eastern Mediterranean, consistent with the intensification of eastward pathways since 1980 reported by Adame et al. (2022). Notably, a March 2003 North African dust event traversed continental scales, depositing 50% of Japan's dust load within a week (Tanaka et al., 2005). Moreover, unlike the westward pathway, LSC-linked dust can be transported eastward across the North Pacific along a considerably longer path, reaching the southeastern region of North America in the second period (Fig. 3b and 3j). The regression analysis of dust aerosol optical depth (DOD) onto LSC from the CMIP6 model reveals that a statistically significant positive correlation ($p < 0.1$) exists between LSC and dust aerosol over the Atlantic Ocean adjacent to West Africa (Fig. S2a). Conversely, after late 1990s, significant negative correlations ($p < 0.1$) emerge between LSC and dust aerosol over North Africa, mid-latitude Asia, and southwestern North America (Fig. S2b). These findings provide robust evidence for the interdecadal variability in dust distribution patterns associated with LSC.

During the first period, Atlantic-bound dust transport predominantly occur within the low-to-mid troposphere (850−500 hPa) (Fig. 3c,3e), as evidenced by vertical cross-sections of dust mixing ratios (DMR) in Saharan (Fig. 3i). The second period reveals an elevated dust layer extending to 10 km altitude (Fig. 3i) demonstrating sustained eastward transport—a pattern attributable to springtime North African dust emissions according to satellite-derived analyses (Yang et al., 2022). Significant positive DMR anomalies are observed at 500 hPa across nearly the entire zonal belt at mid-latitudes (Fig. 3f), consistent with the findings of Uno et al. (2009). Their CALIOP observations and transport model simulations suggest that circum-global dust trajectories persist in upper-troposphere for multiple revolutions before deposition. Notably, our analysis identifies stronger DMR anomalies at 500 hPa than at 850 hPa over the North Pacific (Fig. 3d,3f), highlighting mid-tropospheric dominance in trans-Pacific dust transport.

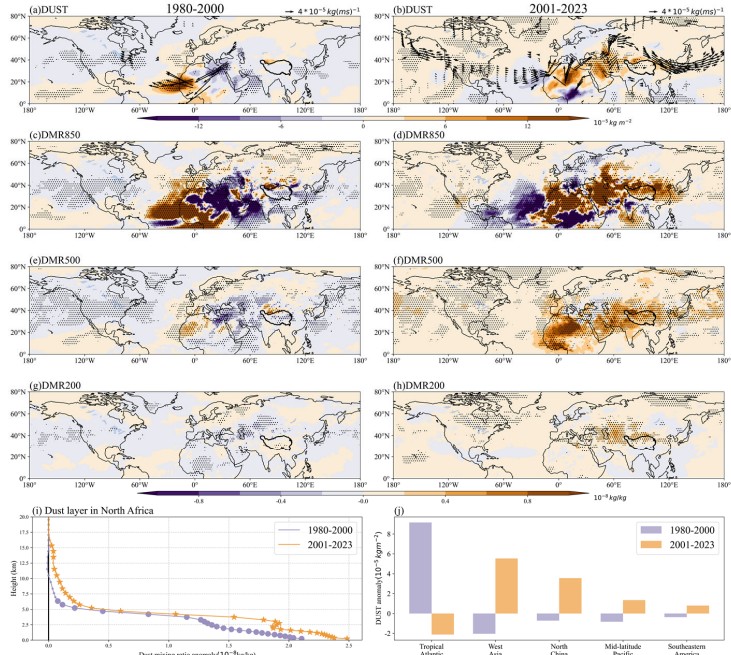

267

**Fig. 3: Changes in spring dust transport pathway and transport height in North Africa associated with pre-winter LSC of composite analysis during the two periods.** Composite analysis of the DUST anomalies (shading; $10^{-5}$kg/m$^2$) and dust column uv-wind mass flux anomalies (vectors; $10^{-5}$kg/ms) in (a) 1980-2000 (positive LSC minus negative LSC) and (b) 2001-2023 (negative LSC minus positive LSC), respectively. (a) Vertical structure of dust mixing ratios anomalies at the North African sand source (5°N−30°N; 18°W−30°E). Spatial characteristics of dust mixing ratios (DMR) anomalies (shading; $10^{-8}$kg/kg) at (c) 850hPa, (e) 500hPa, and (g) 200hPa in 1980−2000 for representative layers. (d) 850hPa, (f) 500hPa, and (h) 850hPa, are for 2001−2023. (i) Vertical structure of DMR anomalies at the North African sand source (5°N−30°N; 18°W−30°E) during the two periods. (j) The regional average of the DUST anomalies in five subregions (West Asia (35°N−40°N; 32°E−60°E), Northern China (34°N−40°N; 95°E−120°E), Mid-latitude Pacific (35°N−50°N; 180°W−150°W.) and Southeastern North America (25°N−40°N; 108°W−83°E). The dashed dots and vectors indicate that the anomalies pass the 90% confidence test.

## 3.2 Mechanisms for the interdecadal shifts of impact of LSC on dust transport path

During the first period, the composite geopotential height anomalies at 500 hPa (Z500) are presented in Fig. 4a and 4c, illustrating the differences between the LSC





positive and negative phases. During the pre-winter period, a general positive anomaly
in Z500 is observed over the NH mid-latitudes, including North America, Eurasia, and
the Atlantic Ocean, while a negative anomaly is evident in higher latitudes (Fig. 4a).
The NAO+ mode is observed in the extratropical Atlantic region, accompanied by PV
anomalies (Fig. 4a, 4g), which typically facilitate downstream Rossby wave breaking,
as reported in previous studies (He et al., 2014; Molteni et al., 2011). In the following
spring, the anomalous anticyclone over northwestern North Africa, triggered by the
winter NAO+ mode, drives anomalous northeasterly winds, transporting dust from the
Sahara Desert to the Atlantic (Fig. 4c, 4h). The 10 m anomalous easterly wind
probability in the tropical Atlantic is significantly higher in the 1980-2000 LSC positive
phase compared to the 2000-2023 LSC negative phase (Fig. 4h, the short red line
represents p < 0.1 in positive LSC). Long-duration, widespread dusty events are
frequently associated with explosive anticyclones situated to the rear of the northern
Sahara Desert (Knippertz et al., 2012). In addition, during positive LSC phase,
weakened and poleward mid-latitude westerlies further amplify terrestrial warming
through a positive feedback mechanism (He et al., 2014), which also enhances dust
activity.
From 2001 to 2023, the Z500 field pattern is completely opposite to that of the first
phase (Fig. 4b), which is considered as the expected outcome of WOCL mode during
winter. However, the lag effect of the pre-winter LSC signal in the subsequent spring
differs from that in the first phase, likely due to interdecadal variability of heat retention
in the ocean memory (Pan et al., 2005; Yu et al., 2024; Khatri et al., 2024; Han and Wu,
2025). Specifically, the strengthening of the tri-polar pattern of sea surface temperatures
anomaly (SSTA) from DJF to MAM (Fig. S3b, S3d) leads to the maintenance of the
NAO− pattern into the spring (Fig. 4d, 4g). The anomalous cyclonic circulation in the
Atlantic strengthens the southwesterly (Fig. 4d and 4i), which direct dust plumes toward
the Eurasian border region. For example, the NAO− phase in March 2018 caused
surface dust concentrations in the eastern Mediterranean to be approximately 200 µg/m³
higher than the climatological value, due to strong southwesterly (Kaskaoutis et al.,
2019). The prevailing stronger westerlies continue to transport dust eastward (Fig. 4f
and Fig. S2d and S4). Meanwhile, under the effect of downward momentum, the
probability of westerlies near the surface in these regions increases (Fig. 4g−4l, the
short blue line represents p < 0.1 in negative LSC), leading to dust deposition in
northern China, the Pacific, and the southeastern United States.

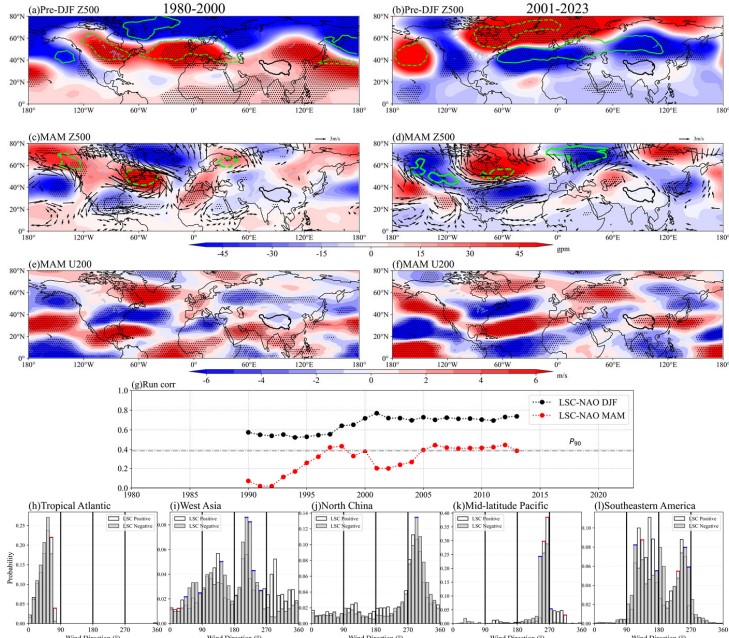

**Fig. 4: Atmospheric anomalies in spring associated with pre-winter LSC of composite analysis during the two periods.** Composite (a) pre-DJF 500 hPa geopotential height (Z500) anomalies (shading; gpm) with 320K potential vorticity (PV320) anomalies (green contours, only absolute values >0.3 are shown, solid lines represent positive, dashed lines represent negative; PVU), (c) MAM Z500 anomalies (shading; gpm) with PV320 and 500hPa horizontal wind (UV500) anomalies (vectors; m/s), and (e) 200hPa U-wind (U200) anomalies (shading; m/s) for 1980−2000 (positive LSC minus negative LSC). Composite (b) pre-DJF Z500 with PV320, (d) MAM500 with PV320 and UV500, and (f) U200, are for 2001−2023 (negative LSC minus positive LSC). (g) The pre-DJF LSC is associated with a 21-year sliding correlation with the NAO over the same period (black line) and in the following spring (red line). The significance at the 90% (gray) levels is shown by the dashed line. Histogram of surface wind directions at all grid points within (h) the Tropical Atlantic during positive LSC (white bars) and during negative LSC (blue bars) in 1980-2000. For surface wind directions, "NE", "SE", "SW", and "NW" indicate north-easterlies, south-easterlies, south-westerlies, and north-westerlies, respectively. (i), (j), (k), and (l) represent the West Asia, northern China, mid-latitude Pacific, and southeastern North America in 2001-2023, respectively. The boxes are filled in red and blue when positive LSC and negative LSC are statistically significant pass the 90% confidence test. The dashed dots indicate that the anomalies pass the 90% confidence test.

The composite analysis from 1980 to 2000 shows that anomalous northeasterly
winds (Fig. 4c) lead to significant positive anomalies wind speed in the western and



central regions of North Africa (Fig. 5a), which align with the spatial distribution of the
second empirical orthogonal function (EOF) of 10m wind speed (Evan et al., 2016).
This wind speed anomaly facilitates dust emission south of 20°N (Fig. 5e), explaining
58% of the variation in westward dust transport across North Africa (Evan et al., 2016).
In contrast, soil conditions exert a smaller influence on dust emission (Fig. 5b). The
cold northeasterly cool the eastern region, triggering anomalous zonal temperature
gradients (Fig. 5c) and alterations in zonal circulation patterns (Fig. 5d). These changes
further amplify the vertical uplift of dust, carrying it into the mid-lower troposphere of
the Atlantic. Additionally, radiative heating effects in the source regions strengthen the
upward motion of dust (Carlson et al., 1980).
From 2000 to 2023, anomalous southwesterly winds cause significant warming in
the northwestern part of North Africa, with maximum anomalies capable of exceeding
4 K (Fig. 5h). This is due to the weakening of the subtropical high (Fig. 4d), which
triggers strong westerly warm advection and enhances vertical mixing in the
atmospheric boundary layer under the NAO− phase (Zhou et al., 2024). The warming
of the surface has two major impacts. First, it intensifies soil drought in the Sahara
Desert (Fig. 5g), weakens the soil cohesion, and promotes dust emissions north of 25°N
(Fig. 5j). Second, the warming strengthens the meridional temperature gradient (Fig.
5h), driving exceptionally strong meridional circulation. This leads to noticeable
upward motion north of 25°N in North Africa, even extending to 200 hPa (Fig. 5i),
generating favorable conditions for dust transport into the upper troposphere, which in
turn facilitates long-range transport eastwards.



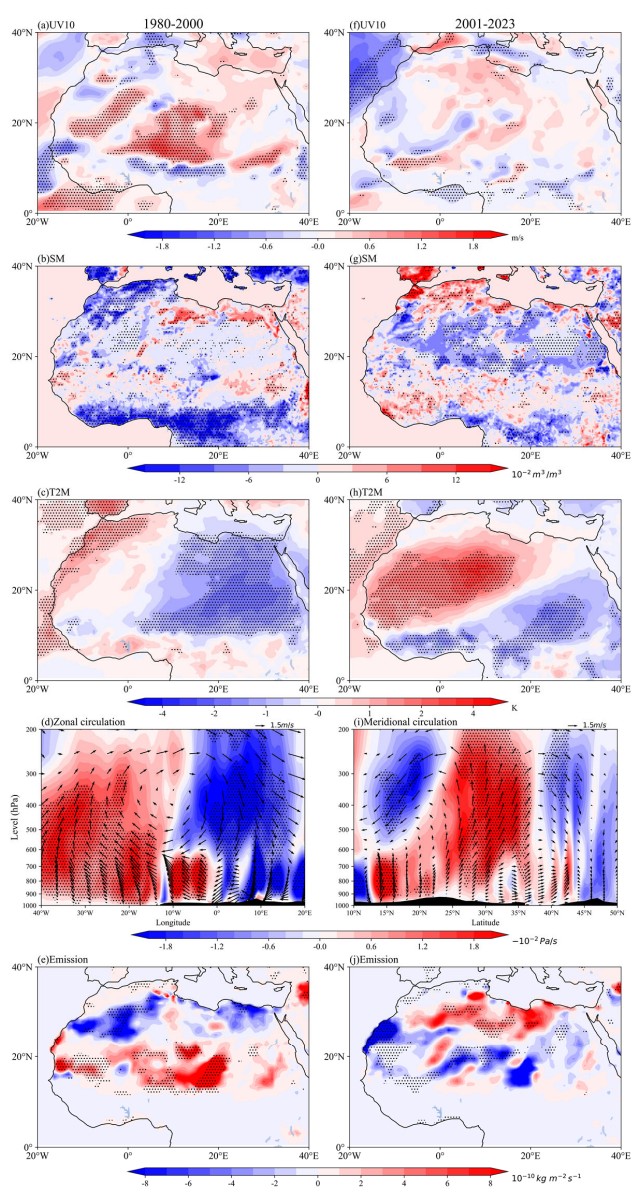





**Fig. 5: Local meteorological conditions and dust emissions during spring in North Africa associated with pre-**
**winter LSC of composite analysis during the two periods.** The composite (positive LSC minus negative LSC)
(a) 10m-wind speed (UV10) anomalies (shading; m/s), (b) soil moisture (SM) anomalies (shading; $10^{-2}$m$^3$/m$^3$), (c)
Two-meter temperature (T2M) anomalies (shading; K), (d) mean cross sections of zonal circulation (vectors; V-wind
for m/s and vertical velocity for Pa/s) anomalies (latitude averaged over 10°N–20°N), and (e)dust emissions
anomalies (shading; $10^{-10}$kgm$^{-2}$s$^{-2}$) in 1980−2000. (g-j) then for the composite (negative LSC minus positive LSC)
2000−2023, where (g) represents the meridional profile averaged over longitudes 20°W-10°E. The shading in (d)
and (i) represents the magnitude of the vertical velocity, which is multiplied by a factor of −150 to enhance the visual
interpretation of wind vectors. The dashed dots indicate that the anomalies pass the 90% confidence test.

## 4 Conclusions and discussions

This study primarily reveals that the dust transport pathway from North Africa in
the subsequent spring, influenced by the preceding winter LSC, shifted from a
westward to a long-range eastward trajectory in the late 1990s. The schematic in Fig. 6
outlines the dynamical processes, ranging from large-scale to local-scale, that control
dust emission, uplift, and subsequent transport. The 1980−2000 LSC+ phase (Fig.6a)
amplifies zonal temperature gradients between warming Eurasian/North American
continents and cooling oceanic basins, driving the NAO+ mode that establishes
intensified anticyclonic systems over northeastern North Africa. These synoptic
configurations generate anomalous northeasterlies that enhance both dust emission and
westward Atlantic transport, corroborated by the dominance of wind-driven emission
mechanisms (Evan et al., 2016). Post-2000, the reversed LSC− phase (Fig. 6b) promotes
NAO− persistence into spring, with anomalous southwesterly advection inducing
Saharan soil desiccation and convective uplift. Midlatitude westerly intensification
enables circum-global dust transport extending to southeastern North America. Overall,
the variation in the LSC-related dust transport directions along the westward and
eastward pathways is closely related to the climatic variability determined by the phases
of the NAO. The significant role of the second dry period on dust emissions, similar to
the findings for the Gobi dust event (Zhu et al., 2024), highlights the significant
influence of regional drought on dust emissions in the context of global warming,
particularly as a consequence of intense heatwaves.

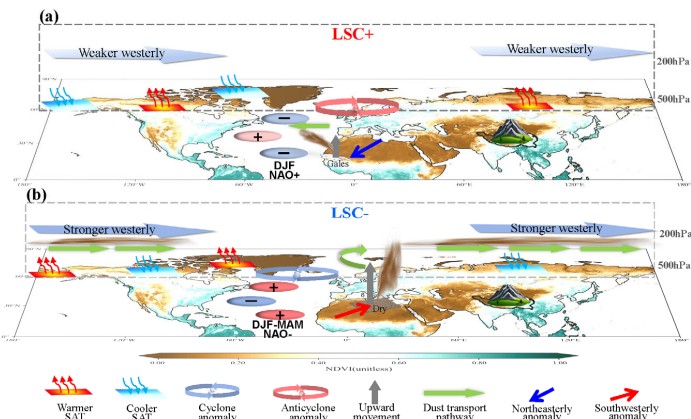

**Fig. 6: A schematic diagram summarizing the sprint dust activity in North African associated with pre-winter LSC over two periods including dust emission, uplift, transport and deposition.** (a) associated with LSC+ in 1980−2000; (b) associated with LSC- in 2000−2023. Here, only the two periods related to positively anomalous dust in North Africa are shown, where dust activity is suppressed when the sign of the LSC in both periods is opposite to that of the schematic diagram. Normalized Difference Vegetation Index (NDVI) values for the bottom graph are from GIMMS ndvi3g (1982−2022).

This study is based on statistical and dynamical diagnostics, with its results can be validated through some other numerical simulations of land-ocean contrasts. Previous research in idealized atmospheric circulation simulations has demonstrated that continental warming (LSC+) induces planetary wave modes, with a dipole resembling NAO emerging as the dominant regional feature (Molteni et al., 2011). This is accompanied by anomalous dispersion of the tropospheric Eliason-Parma fluxes in the mid-latitudes, which reduces the net meridional vortex heat flux into the stratosphere and weakens the westerlies (Portal et al., 2022). This, in turn, supports the conclusions of this paper regarding the eastward transport paths of the second LSC− phase. Additionally, the critical role of NAO-modulated land-atmosphere interactions receive further validation from Sahelian climate studies, where vegetation-precipitation feedbacks amplify dust emission sensitivity to circulation anomalies (Lu et al., 2005; Folland et al., 1986). Our findings align with Global Ozone Chemistry Aerosol Radiation and Transport (GOCART) model simulations that quantitatively link NAO phases to North Atlantic dust load variability (Ginoux et al., 2004), though they extend this paradigm by revealing LSC effects on transcontinental transport efficiency. Moreover, the global signal response is primarily driven by Asian warming, across the zonal boundary region (Portal et al., 2022). This highlights the need for further investigation into the impact of subregional LSC variations on dust transport.



This study elucidates a novel mechanism through which pre-winter land-sea
contrast signals modulate spring dust lifting and transport trans-seasonally, offering
critical insights for advancing the accuracy of dust prediction models. However,
significant non-linear characteristics are exhibited in their dynamics. For example, the
interaction between LSC and NAO (Molteni et al., 2011), the link between NAO and
atmospheric blocking (Athanasiadis et al., 2020; Croci-Maspoli et al., 2007), the
connection between LSC and the westerly jet (He et al., 2014; Portal et al., 2022), and
the coupling between the jet stream and Madden Julian Oscillation (MJO) (Kang et al.,
2018; Bao et al., 2014). These complex nonlinear interactions result in considerable
uncertainties regarding future changes in dust activity (including emission, transport,
and deposition) and their impacts. High sensitivity to the land-ocean boundary response
and to scenarios of future $CO_2$ concentration pathways has been demonstrated in
changes to climate patterns (Kamae et al., 2014). Although it has been predicted in many
studies that the overall trend of global and regional dust may decrease in the future
(Evan et al., 2016; Shao et al., 2013; An et al., 2018; Yang et al., 2020), the long-range
transport of dust and its impacts on climate under the modulation of LSC and its
associated nonlinear dynamical mechanisms remain a critical area requiring further
urgent research.

## Data availability

All datasets utilized in this study are publicly accessible from the following websites: NASA
MERRA-2    dataset    for    aerosol    and    meteorological    products:
https://disc.gsfc.nasa.gov/datasets?page=1&subject=Aerosols&project=MERRA-2;    Met    Office
HadCRUT5    dataset    for    land-sea    surface    temperature:
https://www.metoffice.gov.uk/hadobs/hadcrut5/; Monthly
CMIP6 mode output for aerosol and meteorological products: https://aims2.llnl.gov/search/cmip6/
; NOAA Global Inventory Monitoring and Modeling System (GIMMS) dataset (version number
3g.v1) for Vegetation product: https://daac.ornl.gov/cgi-bin/dsviewer.pl?ds_id=2187.

## Acknowledgements

This research has been supported by the National Key Research & Development (R&D)
Program of China (2019YFA0606801). This research also has been Supported by Supercomputing
Center of Lanzhou University.

## Code availability

The data were analyzed using Python. All relevant codes used in this study are available upon
request from the corresponding author.



## Author contributions

Q.W. conceived the study and performed the analysis under the guidance of Y.L. Y.L. acquired the funding. The first draft of the manuscript was written by Q.W., and all authors commented on the manuscript and contributed to the writing and revising of the paper. All authors read and approved the final manuscript.

## Competing interests

The authors declare that they have no conflict of interest.

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
