# Peer review of "Interdecadal shift in the impact of winter"

_EGUsphere, 2025_

## Author Response (AR1)

**Response to the reviewers of the manuscript**

**"Interdecadal shift in the impact of winter land-sea thermal contrasts on following spring transcontinental dust transport pathways in North Africa"**

**(EGUSPHERE-2025-826)**

Submitted to Atmospheric Chemistry and Physics for publication

21 Feb 2025

**Response to Reviewer#1**

**General comment**

The paper investigates the influence of land-sea thermal contrasts on the dust transport from north Africa and the interdecadal shift of pathways. The topic is interesting and the paper well written and suitable for the Journal. I only have suggestions for a few minor points.

**Response:**

We sincerely appreciate the time and effort you have dedicated to reviewing our manuscript. Your insightful comments and constructive suggestions have been invaluable in enhancing the quality and clarity of our work. Below, we address each of your concerns in detail and hope that our revisions meet your expectations.

**Specific comments:**

**Comment**: Section 2.1.3. Lines 147-151. The choice of these periods is not very clear, and it should be justified. In addition, are the conclusions depending on this choice?

**Response:**

Thanks for your comments. Regarding the justification for period selection (1980-2000 and 2001-2023) in the revised draft, we tested the PC of the dust field from SVD decomposition by a sliding t-test (see new Fig. S1 in Supplement or Fig. 1 of this document) and found a maximum statistical significance climatic abruptness in 2000 ($p<0.05$). This supports the period division (1980–2000 vs. 2001–2023) and aligns with findings from Liu et al. (2023). Therefore, we have added the sliding t-test results as new Fig. S1 in Supplement and emphasized it in the manuscript (L147–150 and L211–212) for the revised version. Moreover, the two periods are also selected to take into account the fact that the composite sample sizes of the two periods are comparable and do not overlap (Table 1).

[Figure]

Fig. 1 Sliding t-test of PC1-DUST in SVD decomposition

To rigorously assess the temporal robustness of our findings, we conducted systematic sensitivity analyses using multiple window lengths (15, 17, 21, and 23 years). The results consistently demonstrate that the PC of the dust field maintains a statistically significant ($p < 0.05$) correlation with the land-sea thermal contrast (LSC) index across all tested temporal scales after the post-2000 period (Fig. 2e, L219). In the revised manuscript (L209-211), we have explicitly stated: "The dust-LSC correlation remains statistically significant regardless of window length selection (Fig. 2e), with particularly stable associations emerging after 2000." These comprehensive analyses confirm that our central conclusions reflect authentic climate-dust interactions rather than methodological artifacts or temporal sampling biases.

**Reference:**

*Liu G, Li J, Ying T. The shift of decadal trend in Middle East dust activities attributed to North Tropical Atlantic variability[J]. Science Bulletin, 2023, 68(13): 1439-1446.*

**Comment**: In my opinion, most of the figures are quite small and not easily readable when printed. The dashed dot vectors are not visible at all.

**Response:**

Thanks for your suggestion. In the revised version, we have made substantial improvements to all figures to ensure optimal clarity and readability. Specifically, we have enlarged all figures (especially Figs. 2-5) to make dashed/dotted vectors and other critical elements more clearly visible. We have carefully adjusted line weights (increasing them by 30-50%) and optimized color schemes to enhance visual contrast. All figures have been regenerated at a high resolution of 800 dpi to meet printing standards

**Comment**: I would eliminate the bold part of each Figure caption.

**Response:**

Thanks for your suggestion. We have revised all figure captions to use standard formatting (no bold text) for consistency with journal style.

**Comment**: Line 294. Why using a threshold of 0.1 for p rather than the usual one of 0.05. Could the conclusions change if a more standard threshold if chosen?

**Response:**

Thanks for your comments. We re-ran all analyses with $p < 0.05$ and found that the primary conclusions remain valid, though some regional details become less pronounced (e.g., the circum-latitude circle propagation in the Asian region, see Fig. 2 of this document).

However, our choice of $p < 0.1$ was carefully considered. Firstly, previous studies on decadal-scale climate variability (e.g., Liu et al., 2023) also maintained a significance of 0.1. Secondly, the slightly more lenient threshold is sometimes warranted, due to the limited sample size of decadal periods and to maintain the continuity of the dust transport path.

[Figure]

Fig. 2 Composite dust column density anomalies (related to LSC) of two periods (1980-2000 and 2001-2023) with different significance levels (0.05 and 0.1).

**Reference:**

*Liu G, Li J, Ying T. The shift of decadal trend in Middle East dust activities attributed to North Tropical Atlantic variability[J]. Science Bulletin, 2023, 68(13): 1439-1446.*

**Response to Reviewer#3**

**General comment**

This study presents a timely and compelling investigation into the interdecadal shift (~late 1990s) in the relationship between winter land-sea thermal contrast (LSC) and spring dust transport pathways from North Africa. The authors identify the North Atlantic Oscillation (NAO) as a key bridge mechanism and demonstrate a transition from westward (pre-2000) to eastward/circum-global (post-2000) dust transport. The findings have implications for dust prediction, climate modeling, and assessing downstream environmental impacts. My specific comments are as follows:

**Response:**

We are deeply grateful for the thoughtful and constructive suggestions you have provided on our manuscript. we provide a point-by-point response to the reviewer's specific comments as follows.

**Specific comments:**

**Comment**: Regarding the choice of 1980–2000 and 2001–2023 periods is based on a sliding t-test, please give period splitting justification. For example, include the sliding t-test figure in supplementary materials to validate the 1990s shift. Alternatively, apply objective change-point detection (e.g., Pettitt test) to strengthen the rationale for the bifurcation.

**Response:**

Thank you for your suggestion. Regarding the period division methodology, we have addressed this concern by conducting a sliding t-test analysis on the principal component (PC) of the dust field derived from SVD decomposition, which identified a maximum statistical significance ($p<0.05$) climate shift around the year 2000 (see new Fig. S1 in Supplement or Fig. 1 of this document). Therefore, the above explanations have added to the "Methods" (L147-150).

We also have added discussion in the revised manuscript (Line 195–196) highlighting the consistency between our findings and previous dust studies (e.g., Liu et al., 2023) that also recognized the early 2000s as a notable climatic transition period. This cross-validation strengthens the rationale for our period selection. We appreciate your suggestion about the Pettitt test and will consider it for future work to further validate the robustness of the detected shift.

[Figure]

Fig. 1 Sliding t-test of PC1-DUST in SVD decomposition

**Reference:**

*Liu G, Li J, Ying T. The shift of decadal trend in Middle East dust activities attributed to North Tropical Atlantic variability[J]. Science Bulletin, 2023, 68(13): 1439-1446.*

**Comment**: The results mention an increase in dust emissions after 2000 due to the drying of soils as a result of anomalous warming (Fig. 5g). However, quantitative analyses (e.g., regression of dust emissions on soil moisture) are not provided in the figure.

**Response:**

Thank you for your suggestion. In our original analysis, we focused primarily on establishing the dynamic linkage between land-sea thermal contrast (LSC) and North African dust variability. We fully acknowledge that the impact of soil moisture changes deserves more explicit quantitative treatment.

As the reviewer pointed out, we have indeed conducted comprehensive composite analyses of LSC onto key dust-emission-related meteorological factors, including 10-m wind speed (shown in Fig. 5a,5f), surface air temperature (shown in Fig. 5c,5h), and soil moisture (shown in Fig. 5b,5g). We regret not highlighting these quantitative results more prominently in our original discussion.

In the revised manuscript (L345-348), we have now explicitly stated: "First, the LSC-induced soil moisture deficit, quantified through composite analysis in Fig. 5g (peak anomalies of 0.03 m³/m³ in 25-30°N), significantly reduces soil cohesion, promoting dust emissions north of 25°N (Fig. 5j)." We thank the reviewer for helping us improve the clarity and completeness of our results.

**Comment**: CMIP6 results (Fig. S2) broadly support observations but lack discussion of model spread/uncertainty. I recommend quantify inter-model agreement (e.g., % of models showing significant correlations) and discuss limitations.

**Response:**

We sincerely appreciate the reviewer's constructive comments regarding the CMIP6 analysis. To address this, we conducted a systematic evaluation using all 14 CMIP6 models with available output variables (DOD and near-surface air temperature). For robust period comparison, we performed separate regressions for 1980–2000 and 1994–2014 (Fig. 2 and 3 of this document). We find four models (including 'CESM2', 'IPSL-CM5A2-INCA', 'BCC-ESM1', and 'CESM2-FV2') showed significant LSC-DOD correlations during 1980–2000, while another four models (including 'CESM2', 'IPSL-CM5A2-INCA', 'MRI-ESM2-0', 'CESM2-WACCM-FV2') exhibited significance during 1994–2014 (with two overlapping models) (Fig. 2 and 3 of this document). Then, the six best-performing models (43% of all) were selected for multi-model averaging (The original Fig. S2 has been changed to Fig. S3 in Supplementary).

In the revised manuscript (L242–249), we explicitly state that by quantifying inter-model agreement, we demonstrate that 43% (6 out of 14) of CMIP6 models reproduce the observed statistically significant spatial correlations between land-sea thermal contrast (LSC) and dust variability. This consensus supports the robustness of the LSC-dust linkage despite known model biases.

We also further clearly discussed the limitations (L404-406): the ensemble member constraint (r1i1p1f1 per model) prevents assessment of intra-model variability. Proposed future work to analyze multi-member ensembles where available.

[Figure]

Fig. 2: Regression patterns of DOD onto LSC in CMIP6 during 1980–2000. (a) CESM2;(b) IPSL-CM5A2-INCA; (c) IPSL-CM6A-LR-INCA; (d) IPSL-CM6A-LR; (e) CanESM5-1; (f) MRI-ESM2-0; (g) MIROC6; (h) CESM2-WACCM; (i) CanESM5; (j) BCC-ESM1; (k) NorESM2-LM; (l) NorESM2-MM; (m) CESM2-WACCM-FV2; (n) CESM2-FV2. The CMIP6 model, including 'CESM2', 'IPSL-CM5A2-INCA', 'MRI-ESM2-0', and 'CESM2-WACCM-FV2', indicates a better simulation of LSC-affected dust in North Africa, with potential transport to the coast of Western Europe. The dashed dots indicate that the anomalies pass the 90% confidence test. All-time series are low-pass filtered to preserve interdecadal variability.

[Figure]

Fig. 3: Regression patterns of DOD onto LSC in CMIP6 during 1994–2014. (a) CESM2;(b) IPSL-CM5A2-INCA; (c) IPSL-CM6A-LR-INCA; (d) IPSL-CM6A-LR; (e) CanESM5-1; (f) MRI-ESM2-0; (g) MIROC6; (h) CESM2-WACCM; (i) CanESM5; (j) BCC-ESM1; (k) NorESM2-LM; (l) NorESM2-MM; (m) CESM2-WACCM-FV2; (n) CESM2-FV2. The CMIP6 model, including 'CESM2', 'IPSL-CM5A2-INCA', 'BCC-ESM1', and 'CESM2-FV2', indicates a better simulation of LSC-affected dust in North Africa and the possibility of its eastward transport. The dashed dots

indicate that the anomalies pass the 90% confidence test. All-time series are low-pass filtered to preserve interdecadal variability.

**Comment**: It is important to emphasize the practical implications of the shift in conclusion.

Response:

Thank you for your suggestion. Our findings reveal that under global warming, pre-winter land-sea thermal contrast (LSC) has emerged as a critical regulator of spring dust transport pathways in North Africa through its modulation of NAO phase ——a discovery with profound practical implications: The post-2000 regime shift to enhanced eastward circum-global transport (versus pre-2000 westward Atlantic export) fundamentally alters dust prediction paradigms, necessitating the incorporation of winter LSC–NAO coupling as a key predictor in operational dust forecasting systems, and phase-dependent adaptation strategies for downwind regions from West Asia to North America now experiencing intensified dust impacts.

In response to this comment, we have further strengthened the Discussion section (L410-419) to explicitly highlight the practical implications of this regime shift for dust prediction and climate adaptation strategies (e.g., transcontinental dust early warning systems, climate model parameterizations, and management of dust-sensitive sectors like aviation and renewable energy.). This LSC–NAO–dust pathway mechanism provides a vital framework for anticipating future transport changes as climate warming continues to alter land-sea thermal gradients.

**Comment**: It is recommended that the text appearing in all figures be formatted consistently (Fig 6) and that the font be made larger.

**Response:**

Thank you for your suggestion. We have now unified all the text formats in the figure and increased the font size to achieve better readability, as shown in Fig. 6 (L382).

**Reference:**

Liu G, Li J, Ying T. The shift of decadal trend in Middle East dust activities attributed to North Tropical Atlantic variability[J]. Science Bulletin, 2023, 68(13): 1439-1446.